# Impact of Q-fever on physical and psychosocial functioning until 8 years after *Coxiella burnetii* infection: An integrative data analysis

**Daphne F. M. Reukers**[1]*, **Cornelia H. M. van Jaarsveld**[1], **Reinier P. Akkermans**[1,2],
**Stephan P. Keijmel**[3], **Gabriella Morroy**[4], **Adriana S. G. van Dam**[4], **Peter C. Wever**[5],
**Cornelia C. H. Wielders**[6], **Koos van der Velden**[1], **Joris A. F. van Loenhout**[1,7],
**Jeannine L. A. Hautvast**[1]

1 Department of Primary and Community Care, Radboud University Medical Center, Radboud Institute for
Health Sciences, Nijmegen, The Netherlands, 2 Radboud University Medical Center, Radboud Institute for
Health Sciences, Scientific Institute for Quality of Care, Nijmegen, The Netherlands, 3 Division of Infectious
Diseases, Department of Internal Medicine, Radboud Expert Centre for Q-fever, Radboud University Medical
Center, Nijmegen, The Netherlands, 4 Department of Infectious Disease Control, Municipal Health Service
Hart voor Brabant, 's-Hertogenbosch, The Netherlands's, 5 Department of Medical Microbiology and
Infection Control, Jeroen Bosch Hospital, 's-Hertogenbosch, The Netherlands, 6 Centre for Infectious
Disease Control, National Institute for Public Health and the Environment, Bilthoven, The Netherlands,
7 Center for Research on the Epidemiology of Disasters (CRED), Institute of Health and Society, Université
catholique de Louvain, Brussels, Belgium

* Daphne.Reukers@radboudumc.nl

org/10.1371/journal.pone.0263239

KINGDOM

**Data Availability Statement:** Data are available
from the Radboud data repository at https://doi.

# Abstract

## Background

This study aimed to determine short- and long-term physical and psychosocial impact of
*Coxiella burnetii* infection in three distinct entities: Q-fever fatigue syndrome (QFS), chronic
Q-fever, and patients with past acute Q-fever without QFS or chronic Q-fever.

## Methods

Integrative data analysis was performed, combining original data from eight studies measur-
ing quality of life (QoL), fatigue, physical and social functioning with identical validated ques-
tionnaires, from three months to eight years after onset infection. Linear trends in each
outcome were compared between Q-fever groups using multilevel linear regression analy-
ses to account for repeated measures within patients.

## Results

Data included 3947 observations of 2313 individual patients (228 QFS, 135 chronic Q-fever
and 1950 patients with past acute Q-fever). In the first years following infection, physical and
psychosocial impact was highest among QFS patients, and remained high without signifi-
cant improvements over time. In chronic Q-fever patients, QoL and physical functioning
worsened significantly over time. Levels of fatigue and social participation in patients with
past acute Q-fever improved significantly over time.

org/10.17026/dans-zpa-fkph (DOI: 10.17026/dans-zpa-fkph).

**Funding:** This work was supported by Q-support [grant number AMPHI150114-00]. Funder had no involvement in study design; data collection, analysis or interpretation; in the writing of the report; or in the decision to submit the article for publication.

**Competing interests:** The authors have declared that no competing interests exist.

## Conclusion

The impact differs greatly between the three Q-fever groups. It is important that physicians are aware of these differences, in order to provide relevant care for each patient group.

## Introduction

Q-fever is a zoonosis caused by the bacterium *Coxiella burnetii*. Most commonly, humans get infected by inhaling aerosol particles from infected animals, such as goats or sheep [1]. Approximately 40% of infected individuals develop symptoms, usually mild and flu-like [1]. Following a *C. burnetii* infection, approximately 1–5% of patients develop persistent infection, also known as chronic Q fever. This mainly manifests as endocarditis or vascular infection in patients with known risk factors, such as pre-existing cardiac valvulopathies, vascular abnormalities, or immunosuppression [1, 2]. Furthermore, it is estimated that around 20% of symptomatic acute Q-fever patients develop Q-fever fatigue syndrome (QFS), which consists of severe debilitating fatigue as well as other symptoms [3, 4].

The largest Q-fever outbreak to date took place in the Netherlands between 2007 and 2010 [5], with in total 4026 notified cases [6]. Following the outbreak, the number of notifications decreased to on average 14 per year in the last three years (2018–2020) [7]. Before this outbreak, several international studies had determined an impact of acute Q-fever up to 10 years after onset of Q-fever, especially on the level of fatigue [8–10]. During and after the Dutch outbreak, additional studies in larger sample sizes were performed measuring the impact of Q-fever on a variety of outcome measures, such as quality of life (QoL), depression, fatigue, social participation, and physical and cognitive functioning [11–21]. These studies confirmed that Q-fever had a significant impact on the level of fatigue and QoL measured in the first 3 months up to 26 months after onset of Q-fever compared to either controls or normative data [11–16]. Studies also showed that Q-fever patients still experienced a significant impact on fatigue and QoL 4 years after onset of Q-fever compared to normative data or a healthy control group [14, 18]. Other studies showed significant improvements over time after a follow-up of 24 months and even absence of significant differences on the level of fatigue and QoL between persons with or without a history of *C. burnetii* infection up to 3–7 years after onset of Q-fever [12, 17]. Studies specifically performed in patients diagnosed with QFS or chronic Q-fever did show significant impact on psychosocial functioning on the long-term, up to 9 years after acute infection [19–21]. Until now, a comprehensive description of the impact of Q-fever on patients, including short- as well as long-term impact and comparing all of the different Q-fever patient groups, is lacking.

This study aimed to gain insight in the impact of Q-fever on physical and psychosocial functioning over time and compare the impact between patients with QFS, chronic Q-fever and patients with past acute Q-fever (without one of the aforementioned diagnoses) by performing an integrative data analysis, i.e. the analysis of original data pooled from multiple studies [22].

## Methods

An integrative data analysis (IDA) was performed on original data from studies measuring physical or psychosocial functioning of Q-fever patients since 2007 in the Netherlands. IDA is the statistical analysis of a single data set that consists of two or more separate samples that

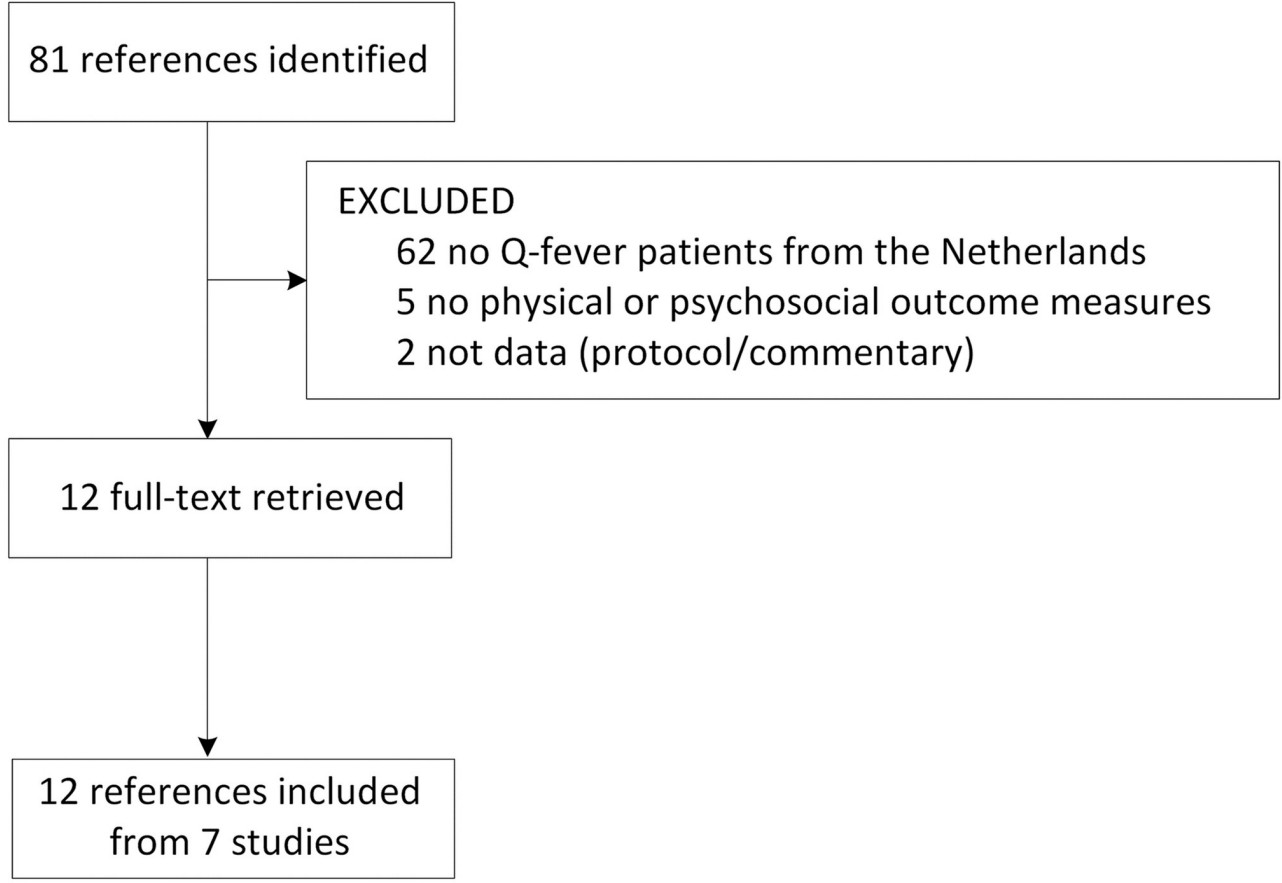

Search terms used (March 2018): (Q-fever[Title/Abstract] OR acute Q-fever[Title/Abstract] OR chronic Q-fever[Title/Abstract] OR Q-fever fatigue syndrome[Title/Abstract] OR QFS[Title/Abstract]) AND (quality of life[Title/Abstract] OR fatigue[Title/Abstract] OR psychosocial[Title/Abstract] OR health status[Title/Abstract] OR physical[Title/Abstract] OR social[Title/Abstract] OR NCSI[Title/Abstract] OR SF-36[Title/Abstract])

**Fig 1. Flowchart of included studies from literature search.**

have been pooled into one [22]. An advantage of IDA is the increase of statistical power as the analysis is performed on a larger sample size and a greater sample heterogeneity resulting in improved generalizability [22]. By combining studies with observations at different timepoints since infection, we were able to perform longitudinal analysis, which is another major advantage.

### Study selection

A literature search in PubMed was performed to identify relevant studies (Fig 1). All studies performed in the Netherlands with Q-fever patients since the start of the epidemic in 2007 and measuring any physical or psychosocial outcome measure with validated questionnaires were eligible to be included in this study. Furthermore, by contacting all researchers from the Dutch national Q-fever research network, unpublished psychosocial data were included.

### Data collection

After identifying all relevant studies, the corresponding researchers were contacted to request the original data from their study for the purpose of this research. When the corresponding

researcher agreed, the original data from their study were sent to the research team using separate files for research data and identifiable patient data. Personal data (full name, gender and date of birth) of patients who participated in multiple studies were used to create a single identifier for every patient. Patients who participated in multiple studies were identified through a trusted third party procedure, which ensured privacy protection. This procedure was approved by the Medical Ethical Review Board of the region Arnhem-Nijmegen (2015–2116). The outcome measures from every patient who participated in one or multiple studies were merged into one dataset.

## Study population

Chronic Q-fever patients were diagnosed by their physician according to the Dutch consensus guideline on chronic Q-fever, which includes laboratory diagnostics, medical examination and radiological imaging findings [23]. QFS patients were diagnosed by their physician according to the criteria as defined in the Dutch multidisciplinary guideline concerning fatigue following an acute Q-fever infection, which includes severe fatigue for at least six months, causing significant disabilities in daily functioning, not being caused by chronic Q-fever or other somatic or psychiatric morbidity, directly related to an acute Q-fever infection, and the fatigue should have been either absent before or have significantly increased since the acute Q fever infection [24]. These diagnoses were registered in the original research databases. Some Q-fever patients were diagnosed with chronic Q-fever or QFS after participation in one or several studies in the first years after acute infection. These patients were then retrospectively classified as either chronic Q-fever or QFS patient in all studies. As one of the criteria is that QFS patients should have had severe fatigue for at least 6 months following acute Q-fever, the psychosocial and physical measurements in the first six months were always performed before a diagnosis QFS could have been made [24]. Similarly, initial measurements in chronic Q-fever patients were likely to be performed before diagnosis as well. In addition to classifying patients as QFS or chronic Q-fever, remaining patients in the database were classified as past acute Q-fever patients. This group consisted of patients who had, based on information from the original databases, a past symptomatic *C. burnetii* infection, but were not diagnosed with either chronic Q-fever or QFS.

The date of the initial *C. burnetii* infection was needed to calculate the time since infection for every observation. A date of onset of infection was based on the date of notification or the inclusion criteria as registered in the original database, and was reported for most study participants (93%). As chronic Q-fever can also develop after an asymptomatic infection, it is understandable that the onset of *C. burnetii* infection was not reported for many of these patients (38.5%) [25]. If the date of onset was not registered, the mean date of onset of all Q-fever patients for whom it was available (i.e. August, 2009) was imputed for that patient. Subsequently, for each observation the time since infection was calculated. Nine time points were identified: baseline (including all data measured at 0 to 6 months after infection), 1 year (measured at 6 months to 1.5 years after infection), 2 years (measured at 1.5 to 2.5 years after infection), etc. up to time point 8 years (measured at 7.5 years to 9 years after infection). Every observation was assigned to one of these time points.

## Outcome measures

The following self-reported outcome measures were included in the analysis; 1) Fatigue, as measured with the Checklist Individual Strength Fatigue (CIS-Fatigue) [26, 27]. An increase in the level of fatigue means more fatigue, i.e. deterioration and a score higher than 34 is considered impaired. 2) QoL, as defined by the Nijmegen Clinical Screening Instrument (NCSI) by

combining the standardized z-scores of two instruments measuring depression and satisfaction with life [28, 29]. An increase means higher QoL, i.e. improvement and a score lower than 86.67 is considered impaired. 3) Physical impairment, as measured with the Sickness Impact Profile (SIP) [30, 31]. An increase means more physical impairment, i.e. deterioration and a score higher than 17.38 is considered impaired. 4) Social participation, as measured with the sub domain 'social functioning' from the Short Form 36 (SF-36) [32]. An increase means more social participation, i.e. improvement, but does not have a cut-off value indicating impairment.

## Statistical analysis

Domain scores of every outcome measure were newly calculated based on original items. The data were merged from different original datasets and not every outcome was measured in every study. Missing data within studies were not imputed, as most studies had a maximum of one percent missing values (Q-Herpen had a maximum of three percent missing values and Snel-Q eight percent). Most of the missing values occurred in the Q-Quest I study, with fifteen percent missing. However, these missing values appeared at random when comparing the mean scores of the previous longitudinal outcome from persons with a missing outcome with persons without missing outcomes. There was also no difference in patient characteristics such as age and gender between people with missing values and people without missing values.

The three Q-fever groups were compared on several characteristics (age at *C. burnetii* infection, gender and education level) with either an ANOVA test (continuous variables) or Chi-square test (categorical variables). In order to analyse the general trends over time in each Q-fever group, a multilevel analysis was performed to account for repeated measures within patients. Models with a random intercept and a random trend over time were used. The impact over time for every Q-fever group was analysed using a two-level multilevel linear regression model, with observation-level data as level 1 and participant-level data as level 2. A linear relation between time and outcome was assumed as, for example, the level of fatigue is expected to gradually increase or decrease over time and not exponentially. The score at baseline (i.e. intercept) was compared between Q-fever groups in order to determine whether the level of each outcome measure at baseline was significantly different between groups. Furthermore, it was tested whether the increase or decrease in the level of each outcome measure per time point (i.e. slope) was different between Q-fever groups by including an interaction term group by time in the model. The model was only corrected for gender, age or education level, if it improved the goodness of fit as determined by the likelihood ratio test. A p-value of $< 0.05$ was considered to be statistically significant, based on two sided tests. The analyses were performed using SAS version 9.2.

## Sensitivity analysis

To assess the sensitivity of our basic multilevel model, two different models were analysed and compared to the basic multilevel model including all data. First, in longitudinal analyses the first and last observation may have a large influence on the results. Therefore, the model was recalculated excluding the first and last observations, i.e. all observations from baseline and 8 years after infection. Second, as time point estimations based on few observations in specific Q-fever groups are less reliable, the model was recalculated only including time points with more than 10 observations per Q-fever group. When one of the Q-fever groups had less than 10 observations at a specific time point for a specific domain, all observations from all Q-fever groups were excluded, thereby excluding all observations from time points baseline, 3, 5 and 8 years after infection.

# Results

## Study selection

Fig 1 shows the study inclusion from our literature search: 81 references were identified of which 69 were excluded for various reasons (see Fig 1 for details). In total, 12 full-text articles were retrieved covering 7 unique studies. Two studies (Q-HORT and ImpaQt), of which the psychosocial data was not (yet) published at the time of the literature search (March 2018), were additionally retrieved through the Q-fever research network. Table 1 shows the study populations, designs, year of onset of infection and included validated questionnaires for each study. Data collections ranged from 3 months up to 9 years after *C. burnetii* infection. Including the unpublished studies, 9 separate Q-fever studies were identified of which 8 provided data and were included in the analysis.

## Study population

The database contained 3947 observations of 2313 individual Q-fever patients, divided in 228 QFS patients, 135 chronic Q-fever patients, and 1950 patients with past acute Q-fever. Fig 2 shows the number of observations per Q-fever group and time point (since *C. burnetii* infection). The largest number of observations was collected at 4 years (n = 1607) after infection and the smallest number of observations at 8 years (n = 57, of which 7 were measured at 9 years) after infection. Patient characteristics are presented in Table 2. A significant difference in average age between the three groups was found. Chronic Q-fever patients were on average older (63 years) at onset of infection, while QFS patients were on average younger (40 years) than patients with past acute Q-fever (49 years). The distribution of males and females was significantly different between the three groups. Chronic Q-fever patients were more often male (78.5%) and QFS patients less often male (46.5%) compared to patients with past acute Q-fever (56.3%). Chronic Q-fever and patients with past acute Q-fever had more often a low education (51.1% and 48.6%, respectively) compared to QFS patients (24.0%).

## Multilevel analyses

The results (mean and 95% confidence intervals) as estimated by the multilevel model for each outcome measure in each Q-fever group are presented in Table 3 and visualised in Fig 3. The mean level of fatigue at baseline for QFS patients was significantly higher compared to chronic Q-fever patients and patients with past acute Q-fever (45.8 vs 35.6 and 37.1, Table 3). There was no significant difference in baseline levels of fatigue between chronic Q-fever patients and patients with past acute Q-fever. Over time, the mean level of fatigue for QFS and chronic Q-fever patients did not significantly change (see also Fig 3A). The mean level of fatigue significantly decreased in patients with past acute Q-fever over time (-0.91 point/year (95%CI: -1.23 to -0.59)), and this change in fatigue was significantly different from the QFS and chronic Q-fever patients.

Similarly, the mean level of QoL at baseline for QFS patients was significantly lower compared to chronic Q-fever patients and patients with past acute Q-fever (72.2 vs 87.4 and 84.9 respectively). The mean level of QoL did not significantly change over time for QFS and patients with past acute Q-fever (see also Fig 3B), but significantly decreased for chronic Q-fever patients (-1.4 point/year (95%CI: -2.37 to -0.43)). The change in QoL in chronic Q-fever patients was also significantly different from QFS and patients with past acute Q-fever.

The mean level of physical impairment at baseline for QFS patients was higher compared to patients with past acute Q-fever (13.9 vs 8.4), and the level for chronic Q-fever patients overlapped with both QFS patients and patients with past acute Q-fever (12.6, see Table 3). The

**Table 1. Overview of Q-fever studies eligible for the integrative data analysis.**

| Study name | Corresponding author, year of publication(s) | Study population | ref | N | Study design | Time points of outcome assessment | Outcome measures | | | |
|---|---|---|---|---|---|---|---|---|---|---|
| | | | | | | | Fatigue (CIS) | Quality of Life [1] (BDI/ SWL) | Physical impairment (SIP) | Social participation (SF-36) |
| Case-Control | Limonard, 2010, 2016 [2] | Q-fever patients from Herpen notified in 2007 | [16] | 54 | Cross-sectional | 1 year after infection | x | x | x | |
| | | | [18] | 46 | | 4 years after infection | x | x | x | |
| Q-Quest I | Morroy, 2011 | Q-fever patients notified in 2007–2008 | [15] | 515 | Prospective Cohort | between 12 to 26 months after infection | x | x | x | |
| Q-Quest II | van Loenhout, 2012, 2013, 2014, 2015 | 1. Q-fever patients notified in 2007–2008 and non-notified laboratory confirmed Q-fever patients from 2008–2009 | [14] | 448–193 | Cross-sectional | 4 years after infection | x | x | x | |
| | | 2. Q-fever patients notified in 2010–2011 | | 336 | Prospective Cohort | at 3, 6, 9, 12, 18 and 24 months after infection | x | x | x | x |
| Q-HORT | Wielders, 2015 [3] | Q-fever patients not included in Q-Quest I notified in 2007–2008 and laboratory confirmed chronic Q-fever patients | [33] [3] | 871 | Cross-sectional | 4 years after infection | x | | | |
| Snel-Q | van Dam, 2015 | Q-fever patients with lower respiratory tract infection in general practice in 2009 | [11] | 50 | Cross-sectional | 1 year after infection | x | x | x | |
| QAAD | Hagenaars, 2015 | Vascular chronic Q-fever patients | [20] | 26 | Prospective Cohort | at 3, 6, 9, 12, 15 and 18 months after diagnosis of chronic Q-fever | | | | x |
| Q-Herpen | Morroy, 2016 | Laboratory confirmed acute Q-fever including confirmed chronic Q-fever | [17] | 510 | Cross-sectional | between 3 to 7 years after infection | x | x | x | |
| Qure | Keijmel, 2017 | Q-fever fatigue syndrome (QFS) patients | [21] | 154 | Randomized controlled trial (RCT) | between 1 to 8 years after infection | x | | x | x |
| ImpaQt | Reukers, 2019 [4] | Chronic Q-fever patients | [19] [4] | 80 | Cross-sectional | between 5 to 9 years after infection | x | x | x | x |
| | | QFS patients | | 155 | | | | | | |

[1] BDI and SWL combined form the Quality of Life scale from the Nijmegen Clinical Screening Instrument (NCSI).

[2] This study was identified through the literature search, but not included in the integrative data analysis.

[3] This study was published, but this publication did not include results on the physical or psychosocial outcome measures. Therefore, this publication was not identified through the literature search [33].

[4] This study was published after the literature search was performed [19].

N = number of participating patients; QFS = Q-fever fatigue syndrome; CIS-fatigue = checklist individual strength fatigue; BDI = Beck's depression index;

SWL = satisfaction with life questionnaire; SIP = sickness impact profile; SF-36 Social = short form 36 social functioning.

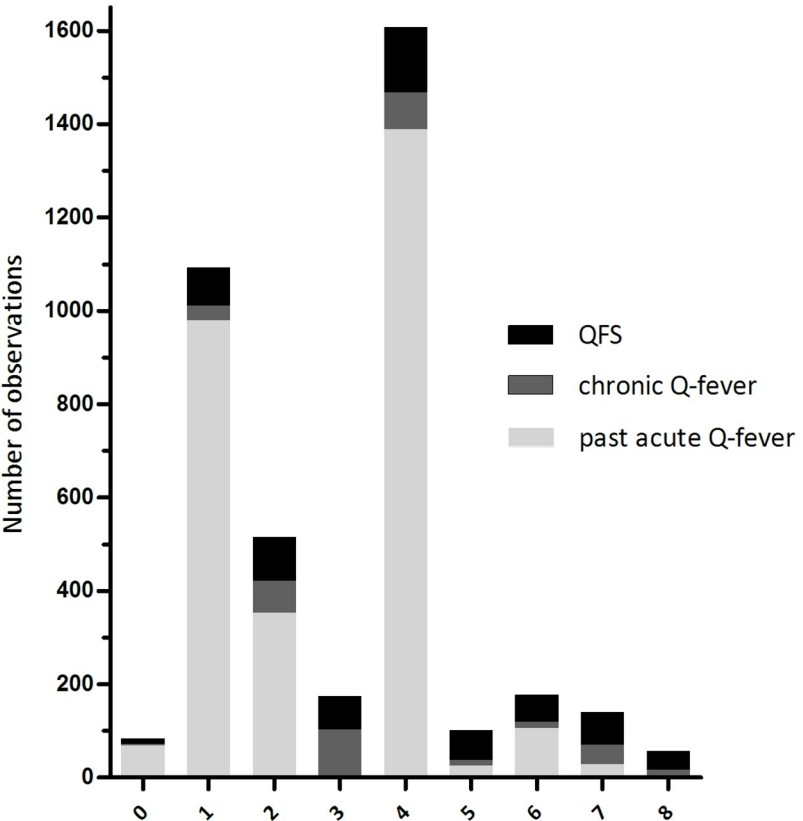

**Fig 2. Number of observations per Q-fever group and time point.** QFS = Q-fever fatigue syndrome.

mean level of physical impairment increased significantly over time for QFS (0.51 point/year (95%CI: 0.14 to 0.87)) and chronic Q-fever patients (1.28 point/year (95%CI: 0.54 to 2.02)), but not for patients with past acute Q-fever (-0.06 point/year (95%CI: -0.32 to 0.19)) (see also Fig 3C). This effect in patients with past acute Q-fever was also significantly different compared to QFS and chronic Q-fever patients.

**Table 2. Description of study populations.**

|  |  | QFS | Chronic Q-fever | past acute Q-fever | p-value | Total |
|---|---|---|---|---|---|---|
| **N patients** |  | 228 | 135 | 1950 |  | 2313 |
| **N observations** |  | 631 | 362 | 2954 |  | 3947 |
| **Age at *C. burnetii* infection** | mean (sd) | 40 (12.2) | 63 (11.3) | 49 (13.6) | <.001 | 49 (14.1) |
| **Gender (male)** | % | 46.5 | 78.5 | 56.3 | <.001 | 56.6 |
| **Education level** |  |  |  |  |  |  |
| **low** | % | 24.0 | 51.1 | 47.3 | <.001 | 44.9 |
| **moderate** | % | 44.3 | 28.9 | 28.1 |  | 29.9 |
| **high** | % | 31.7 | 20.0 | 24.6 |  | 25.2 |
| **missing** | n | 7 | 45 | 273 |  | 325 |
| **Onset *C. burnetii* infection reported** | % | 99.1 | 61.5 | 94.9 |  | 93.3 |

QFS = Q-fever fatigue syndrome; sd = standard deviation.

**Table 3. Results (intercept and slope) from the multilevel linear regression model by Q-fever group corrected for gender.**

| | Intercept: score at baseline | | | Slope: change in score (per year) | | |
|---|---|---|---|---|---|---|
| | QFS | Chronic Q-fever | past acute Q-fever | QFS | Chronic Q-fever | past acute Q-fever |
| | β (95% CI) | β (95% CI) | β (95% CI) | β (95% CI) | β (95% CI) | β (95% CI) |
| Fatigue[1] | 45.8 (43.3; 48.3) [a] | 35.6 (30.5; 40.8) [b] | 37.1 (35.8; 38.3) [b] | -0.18 (-0.75; 0.39) [a] | 0.52 (-0.50; 1.53) [a] | -0.91 (-1.23; -0.59) [b] * |
| Quality of life[2] | 72.2 (68.5; 75.9) [a] | 87.4 (81.5; 93.3) [b] | 84.9 (83.5; 86.3) [b] | 0.28 (-0.34; 0.90) [a] | -1.4 (-2.37; -0.43) [b] * | 0.12 (-0.19; 0.43) [a] |
| Physical impairment[1] | 13.9 (12.0; 15.8) [a] | 12.6 (8.2; 16.9) [a,b] | 8.4 (7.3; 9.4) [b] | 0.51 (0.14; 0.87) [a] * | 1.28 (0.54; 2.02) [a] * | -0.06 (-0.32; 0.19) [b] |
| Social participation[2] | 50.5 (43.1; 57.8) [a] | 44.6 (36.1; 53.0) [a] | 68.5 (64.5; 72.5) [b] | 0.12 (-1.20; 1.40) [a] | 1.10 (-0.54; 2.80) [a] | 5.20 (3.30; 7.20) [b] * |

[a,b,c] The same superscript letter in each row denotes which intercept (score at baseline) does not differ significantly between Q-fever groups, based on overlapping 95% CI, and which slope (score per time point) does not differ significantly between Q-fever groups by testing the significance of fixed effects at the 0.05 level. Consequently, different letters represent significant differences.

* Change in slope (score per year) is significant at the 0.05 level (every intercept (score at baseline) is significant at the 0.05 level).

[1] Higher scores mean higher levels of fatigue; more physical impairment. A positive slope indicates deterioration, i.e. an increase in levels of fatigue, or an increase in physical impairment.

[2] Higher scores mean higher levels (better) quality of life or social participation. A positive slope indicates an improvement, i.e. an increase in quality of life, or an increase in social participation.

QFS = Q-fever fatigue syndrome; CI = confidence interval.

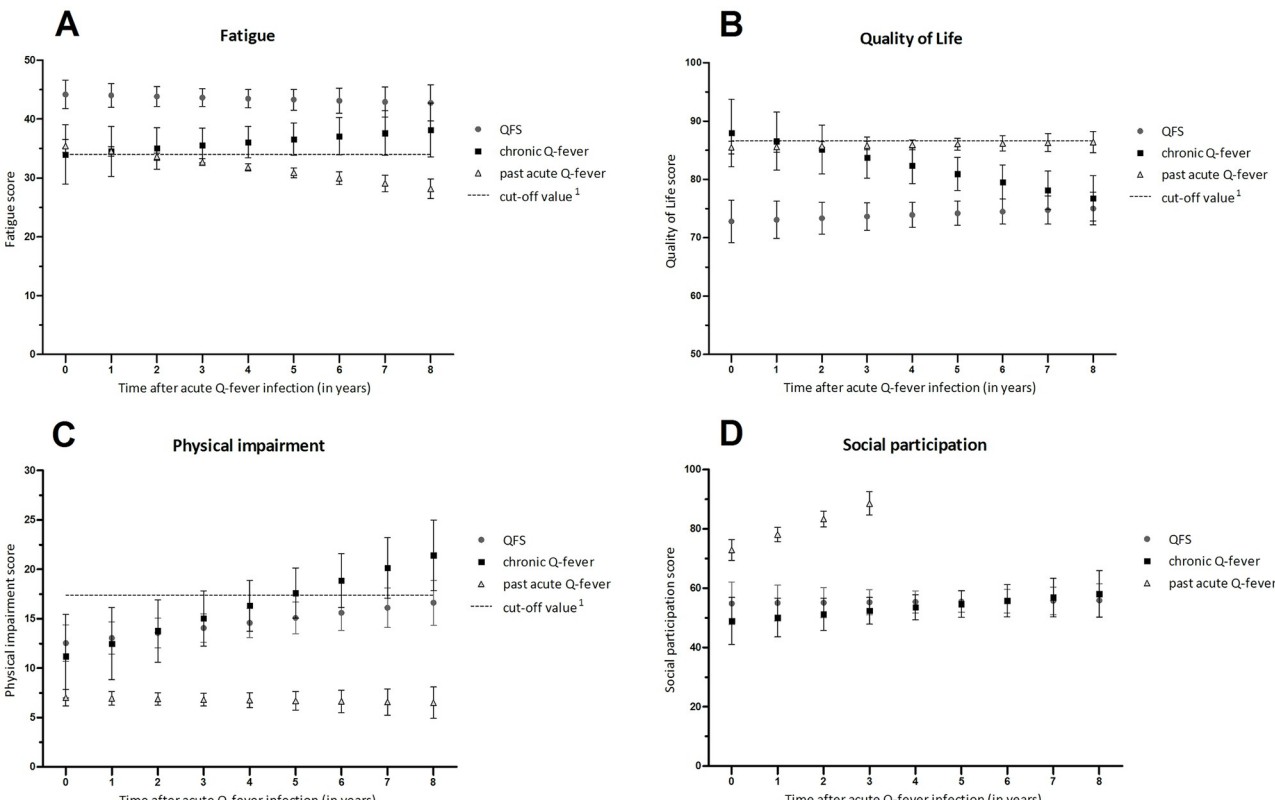

**Fig 3. Error bars representing the estimated mean and 95% confidence interval (corrected for gender) of every outcome measure (fatigue, quality of life, physical impairment and social participation) on every time point since acute Q-fever infection separate for every Q-fever group.**
[1]Participants with a score higher (fatigue, physical impairment) or lower (Quality of Life) than the cut-off value are classified as 'impaired' in that specific domain. QFS = Q-fever fatigue syndrome.

At baseline, patients with past acute Q-fever showed a higher level of social participation than QFS and chronic Q-fever patients (68.5 vs 50.5 and 44.6, respectively). Also, over time, the mean level of social participation significantly increased for this group (5.20 point/year (95%CI: 3.30 to 7.20)), while for QFS and chronic Q-fever patients it remained constant over time. This effect in in patients with past acute Q-fever was significantly different from QFS and chronic Q-fever patients. In patients with past acute Q-fever, social participation was not measured beyond 4 years after infection in any of the individual studies. As there was no data available to support a long-term estimate, the mean and 95% confidence interval were not calculated beyond this time point for this Q-fever group (see also Fig 3D).

### Sensitivity analysis

The first (excluding all observations from time-points baseline and 8 years after infection) and second (excluding all observations from time points baseline, 3, 5 and 8 years after infection) sensitivity analyses showed no major differences compared to the basic model in any of the outcome measures (Table 4 in S1 Appendix).

## Discussion

To our knowledge, this is the largest study on the short- and long-term physical and psychosocial impact of Q-fever ever performed and the first study comparing three groups of distinct entities following a *C. burnetii* infection. This study showed that QFS patients report significantly more fatigue, more physical impairment, lower QoL, and lower social participation than patients with past acute Q-fever without the development of QFS or chronic Q-fever in the first six months after *C. burnetii* infection. They also report significantly more fatigue and lower QoL than chronic Q-fever patients in the first six months after *C. Burnetii* infection. Furthermore, QFS patients showed no improvement in any of the measures and even showed an increase in physical impairment over time. In the first six months following *C. burnetii* infection, chronic Q-fever patients do not show a lower physical or psychosocial functioning compared to patients with past acute Q-fever, except for social participation. However, levels of physical impairment and QoL deteriorate significantly over time in chronic Q-fever patients. Furthermore, the largest group, patients with past acute Q-fever, shows significant improvements in functioning over time.

The finding that QFS patients experience more impact on physical and psychosocial functioning in the first six months after infection than both chronic Q-fever patients and patients with past acute Q-fever has not been previously reported. In addition, our results demonstrated no significant improvements over time in QFS patients, despite the fact that some of these patients had received treatment for QFS [21]. Although this contrasts the finding that cognitive behavioural therapy (CBT) was effective in reducing severe fatigue [21], it is now known that this initial beneficial effect diminishes in the long-term, and therefore corresponds with these findings. However, the design of our study and the fact that no data were available on attended treatments prevented us from determining any further treatment effects. The stagnation of physical and psychosocial functioning in QFS patients shown in our study was comparable with studies in patients with chronic fatigue syndrome (CFS). A review on CFS patients showed that a small improvement in symptoms is commonly reported, but full recovery is very rare [34].

Our study demonstrated that the physical and psychosocial functioning deteriorates over time for chronic Q-fever patients. As chronic Q-fever is a life threatening illness with severe complications, such as acute aneurysm, heart failure or non-cardiac abscesses, which occur in more than 60% of proven chronic Q-fever patients over time, this finding is not surprising

[35]. Furthermore, it has been shown that treatment for chronic Q-fever, which consists of a combination of antibiotics often causing serious side-effects, has a negative effect on QoL [36].

The largest group, patients with past acute Q-fever without subsequent development of QFS or chronic Q-fever, showed a higher physical and psychosocial functioning in the first six months after infection compared to QFS and chronic Q-fever patients. However, this does not mean their physical and psychosocial functioning was not impaired. To give some indication about recovery or impairment, it is possible to compare the scores to a reference group from the general population as reported in Reukers et al. [19]. In that study, the reference group, which was a general population in the Netherlands, reported mean scores of 86.9 on QoL, 22.6 on fatigue, 4.9 on physical impairment and 84.3 on social participation. Comparing these mean scores to the confidence intervals of the subgroup with past acute Q-fever infections reported in our analysis on different time points (as presented in Fig 3) shows that this subgroup reaches similar scores over time on QoL, physical impairment and social participation, but the level of fatigue is higher compared to the reference group in the study of Reukers et al. [19]. Furthermore, studies in other infectious diseases have shown that patients also experience increased fatigue and reduced QoL after infection. Patients with Lyme borreliosis (LB) commonly report fatigue as a prominent symptom during and after Lyme disease (or resolution of erythema migrans), even years after onset or diagnosis of LB [37]. A study on patients with legionnaires disease (LD) showed that 75% of patients experienced fatigue 17 months after completion of antibiotic treatment and on average LD patients experienced lower health-related QoL compared to age and sex matched controls [38]. Our study did show that the physical and psychosocial functioning of patients with past acute Q-fever without subsequent QFS or chronic Q-fever significantly improved over time. Similarly, a study by Wills et al. showed that QoL scores of patients with Lyme disease were below the US national average at baseline, but increased to above the national average after 3 years of follow-up [39].

This study had several strengths. First, this study had a large sample size, which reduced the margin of error from missing data and possible outliers. The sample size for the group of chronic Q-fever patients was relatively small. However, 135 patients is still a sufficiently large study group in order to draw valid conclusions. In addition, since the number of chronic Q-fever patients in our study comprises around 50% of the nationally registered proven chronic Q-fever patients (n = 249), our results represent a large proportion of chronic Q-fever patients in the Netherlands [35]. Also, the number of patients in each Q-fever group is broadly comparable to their prevalence, as approximately 20% of symptomatic acute Q-fever patients develops QFS and chronic Q-fever only develops in 1–5% of either symptomatic or asymptomatic *C. burnetii* infections [2, 4]. Second, this study created a large data set by combining the results of multiple studies performed on different time points using individual patient data. Thereby, we were able to increase the statistical power, perform longitudinal analyses and compare different Q-fever subgroups. Third, by including unpublished studies, possible publication bias was avoided. Fourth, the results were in line with individual studies included in this IDA [11–15, 17, 19–21]. Fifth, all QFS and chronic Q-fever patients included in our analysis were diagnosed in the original studies according to national guidelines.

A limitation of this study were the possible differences in the methods of data collection and the response rates between studies. However, each study used the same validated questionnaires, which were compared between studies. Second, individual studies included in this IDA potentially had participation bias related to outcome measures. It is possible that recovered Q-fever patients may perceive participation not relevant and be less inclined to participate. In addition, severely impaired Q-fever patients might also be underrepresented as they might not prioritize participation in research studies. The results in this study might therefore over- or underestimate the impact of Q-fever. However, this remains speculative, as it was not possible

to determine if and what kind of participation bias occurred in the individual studies. Third, the model assumed a linear disease course, while individual patients may have a fluctuating illness trajectory over time, as shown in patients who developed chronic fatigue syndrome after giardiasis [40]. However, we feel that in this model a linear assumption was most accurate, as on average the patient groups did not show any exponential increase or decrease over time, and we were interested in the trend of the population as a whole. Fourth, the group of patients with past acute Q-fever might have included some undiagnosed QFS or chronic Q-fever patients or were possibly diagnosed with chronic Q-fever or QFS after participation in a study. However, 1381 patients participated in either the Q-HORT or Q-Herpen studies (between 3–7 years after *C. burnetii* infection) and received laboratory diagnostics on chronic Q-fever of which 25 were diagnosed with chronic Q-fever [17, 33]. Patients from the Q-HORT study with high fatigue scores were referred to the Q-fever expert centre to examine whether these patients had QFS. It is also unlikely that this group would include enough undiagnosed QFS or chronic Q-fever patients to have an effect on the outcomes on a sample size of 1950 patients with past acute Q-fever, as both diagnoses are relatively rare.

## Conclusion

QFS patients reported a continuously high physical and psychosocial impact, from the start of the infection onwards. In contrast, the impact for chronic Q-fever patients worsens over time, while the overall majority of patients, belonging to the group of patients with past acute Q-fever, showed significant improvements over time. In conclusion, the physical and psychosocial impact differs greatly between QFS, chronic Q-fever and patients with past acute Q-fever without subsequent development of QFS or chronic Q-fever. It is important that physicians and policy makers are aware of these differences, in order to provide tailored information and offer relevant care for each patient group.

## Supporting information

**S1 Appendix.**
(DOCX)

## Author Contributions

**Conceptualization:** Daphne F. M. Reukers, Cornelia H. M. van Jaarsveld, Koos van der Velden, Joris A. F. van Loenhout, Jeannine L. A. Hautvast.

**Data curation:** Daphne F. M. Reukers, Cornelia H. M. van Jaarsveld, Stephan P. Keijmel, Gabriella Morroy, Adriana S. G. van Dam, Peter C. Wever, Cornelia C. H. Wielders, Joris A. F. van Loenhout.

**Formal analysis:** Daphne F. M. Reukers, Cornelia H. M. van Jaarsveld, Reinier P. Akkermans, Jeannine L. A. Hautvast.

**Funding acquisition:** Koos van der Velden, Joris A. F. van Loenhout, Jeannine L. A. Hautvast.

**Methodology:** Daphne F. M. Reukers, Cornelia H. M. van Jaarsveld, Reinier P. Akkermans, Koos van der Velden, Joris A. F. van Loenhout, Jeannine L. A. Hautvast.

**Project administration:** Daphne F. M. Reukers, Jeannine L. A. Hautvast.

**Supervision:** Cornelia H. M. van Jaarsveld, Koos van der Velden, Joris A. F. van Loenhout, Jeannine L. A. Hautvast.

**Visualization:** Daphne F. M. Reukers.

**Writing – original draft:** Daphne F. M. Reukers, Cornelia H. M. van Jaarsveld.

**Writing – review & editing:** Daphne F. M. Reukers, Cornelia H. M. van Jaarsveld, Reinier P. Akkermans, Stephan P. Keijmel, Gabriella Morroy, Adriana S. G. van Dam, Peter C. Wever, Cornelia C. H. Wielders, Koos van der Velden, Joris A. F. van Loenhout, Jeannine L. A. Hautvast.

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
