## [Decision Letter · Decision Letter 0]

15 Jun 2021

PONE-D-21-13096

Impact of Q-fever on physical and psychosocial functioning until 8 years after Coxiella burnetii infection: an integrative data analysis

PLOS ONE

Dear Dr. Reukers,

Thank you for submitting your manuscript to PLOS ONE. After careful consideration, we feel that it has merit but does not fully meet PLOS ONE’s publication criteria as it currently stands. Therefore, we invite you to submit a revised version of the manuscript that addresses the points raised during the review process.

Many thanks for submitting your manuscript to PLOS One

It was reviewed by two experts in the field, and they have recommended some modifications be made prior to acceptance

I therefore invite you to make these changes and to write a response to reviewers which will expedite revision upon resubmission

I wish you the best of luck with your modifications

Hope you are keeping safe and well in these difficult times

Thanks

Simon

We look forward to receiving your revised manuscript.

Kind regards,

Simon Clegg, PhD

Academic Editor

PLOS ONE

Journal Requirements:

Reviewers' comments:

Reviewer's Responses to Questions

**Comments to the Author**

1. Is the manuscript technically sound, and do the data support the conclusions?

Reviewer #1: Yes

Reviewer #2: Yes

2. Has the statistical analysis been performed appropriately and rigorously? 

Reviewer #1: Yes

Reviewer #2: Yes

3. Have the authors made all data underlying the findings in their manuscript fully available?

Reviewer #1: Yes

Reviewer #2: Yes

4. Is the manuscript presented in an intelligible fashion and written in standard English?

Reviewer #1: Yes

Reviewer #2: Yes

5. Review Comments to the Author

Reviewer #1: The manuscript is very interesting both in terms of sample size and in terms of examining different effects. This study provided valuable information. I think this manuscript can be published with a little change.

1. In the introduction, note how many cases of Q fever are diagnosed and reported annually in the Netherlands (in the last three years)? Please add to t the manuscript.

2. Can the effects of abortion as well as infertility in women be examined and added to the text? I suggest that these items be added to the manuscript.

3. The comparison results of the present study with previous studies should be presented in the form of a table so that the obtained results are well understood.

4. The format of the references does not conform to the journal guidelines. Please correct.

Reviewer #2: Summary:

Data was gathered from several studies comparing Q fever groups in aspects such as fatigue, quality of life, physical impairment, and social participation. Groups included chronic Q fever, acute Q fever, and Q fever fatigue syndrome patients. This data analysis concluded chronic Q fever patients worsened over the years in terms of quality of life and physical impairment whereas acute Q fever patients improved. QFS patients did not improve over time in terms of physical impairment and social participation.

Major Comments:

1. These studies were combined and compared to generate this data set when the data was taken separately (from different groups) with different parameters/methods. Not all data such as fatigue, quality of life, physical impairment, and social participation was researched in each chosen study. This represents a flaw in the compilation method for data analysis.

2. Social participation data points for acute Q fever patients does not span the full study.

Minor Comments:

1. In many places Coxiella burnetii is spelled “Coxiella burnetiid” or “Coxiella Burnetii”.

6. PLOS authors have the option to publish the peer review history of their article (what does this mean?). If published, this will include your full peer review and any attached files.

Reviewer #2: No

---

## [Author Response · Author response to Decision Letter 0]

27 Oct 2021

Reviewer #1:

1. In the introduction, note how many cases of Q fever are diagnosed and reported annually in the Netherlands (in the last three years)? Please add to t the manuscript.

RESPONSE: We have added this information to the introduction (page 3, lines 12-13):

[The largest Q-fever outbreak to date took place in the Netherlands between 2007 and 2010 5, with in total 4026 notified cases]. Following the outbreak, the number of notifications decreased to on average 14 per year in the last three years (2018-2020).

2. Can the effects of abortion as well as infertility in women be examined and added to the text? I suggest that these items be added to the manuscript.

RESPONSE: Some studies have indeed shown that a C. burnetii infection can lead to a higher risk of adverse pregnancy outcomes, such as a higher risk of miscarriage, preterm delivery and low birth weight. However, we have no data available on these outcomes in our study. Therefore, we feel this part of the impact that C. burnetii infection can have is too specific and beyond the scope of our manuscript. 

3. The comparison results of the present study with previous studies should be presented in the form of a table so that the obtained results are well understood.

RESPONSE: It would not be appropriate to compare the results (i.e. the average scores on the different questionnaires) between our results and the individual studies for two reasons. First, the scores reported in our manuscript are estimated by a multilevel linear model. This method was chosen in order to account for repeated measures and to compare the general trends over time between the three Q fever entities, adjusted for gender differences between patient groups. The adjusted scores on any time point as presented in Table 3 and Figure 3 will therefore not accurately represent unadjusted scores on the validated questionnaires and it would not be fitting to compare these estimated scores to the average scores as reported in the individual studies. Second, because we had access to individual patient data, we were able to retrospectively classify patients as QFS or chronic Q fever patients in earlier studies, if they received one of these diagnoses after participation in these studies. However, during participation in these earlier studies, they were still classified as (past) acute Q fever patients. Therefore, the scores of our study and those of the earlier studies, where patients were not yet divided between these three entities, cannot be properly compared. 

4. The format of the references does not conform to the journal guidelines. Please correct.

RESPONSE: We apologize for not properly formatting the references according to the guidelines. We have corrected this in the manuscript.

Reviewer #2:

Major Comments:

1. These studies were combined and compared to generate this data set when the data was taken separately (from different groups) with different parameters/methods. Not all data such as fatigue, quality of life, physical impairment, and social participation was researched in each chosen study. This represents a flaw in the compilation method for data analysis.

RESPONSE: Inclusion of all parameters in each study would have been favourable. Standardisation across studies is of great benefit for meta-analyses and studies like ours. Even though each study included in our analysis did not include all four parameters (fatigue, quality of life, physical impairment, social participation), we feel this did not have a significant impact on the compilation or data analysis. We included 8 studies and the parameters were included in 7, 5, 6 and 4 of the 8 studies, respectively (Table 1). This shows that the analysis on these individual parameters were not based on just 1 or a limited amount of studies, which could cause an imbalance in how central the data of this particular study would be for the results. Also, the multilevel model is very well equipped to handle missing values in the outcomes. 

With regard to this comment, we agree that the parameter social participation might have some limitations, which we will discuss below in the second comment of the reviewer. 

2. Social participation data points for acute Q fever patients does not span the full study.

RESPONSE: Unfortunately, social participation was not included in any of the long-term individual studies in acute Q fever patients. Therefore, we did not feel it was appropriate to estimate the scores for this parameter in acute Q fever patients beyond 4 years after infection, as we did not have any data to support this estimate. This was reported in the results section, however, we have added some additional substantiation in this sentence, to clarify this point (page 12, lines 204-208):

In patients with past acute Q-fever, social participation was not measured beyond 4 years after infection in any of the individual studies. As there was no data available to support a long-term estimate, the mean and 95% confidence interval were not calculated beyond this time point for this Q-fever group. 

Minor Comments:

1. In many places Coxiella burnetii is spelled “Coxiella burnetiid” or “Coxiella Burnetii”.

RESPONSE: We apologize for not properly checking the manuscript for these errors. We have corrected this in the revised manuscript. 

We greatly appreciate your time and effort for reading our revisions and hope that our responses provided above have satisfactorily addressed the questions and comments of the reviewers. 

Kind regards, on behalf of all co-authors,

Daphne F.M. Reukers

---

## [Editor Report · Decision Letter 1]

18 Jan 2022

Impact of Q-fever on physical and psychosocial functioning until 8 years after Coxiella burnetii infection: an integrative data analysis

PONE-D-21-13096R1

Dear Dr. van Jaarsveld,

We’re pleased to inform you that your manuscript has been judged scientifically suitable for publication and will be formally accepted for publication once it meets all outstanding technical requirements.

Kind regards,

Simon Clegg, PhD

Academic Editor

PLOS ONE

Additional Editor Comments:

Many thanks for resubmitting your manuscript to PLOS One

As you have addressed all the comments and the manuscript reads well, I have recommended it for publication

You should hear from the Editorial Office shortly.

It was a pleasure working with you and I wish you the best of luck for your future research

Hope you are keeping safe and well in these difficult times

Thanks

Simon

---

## [Editor Report · Acceptance letter]

20 Jan 2022

PONE-D-21-13096R1 

Impact of Q-fever on physical and psychosocial functioning until 8 years after *Coxiella burnetii* infection: an integrative data analysis 

Dear Dr. van Jaarsveld:

I'm pleased to inform you that your manuscript has been deemed suitable for publication in PLOS ONE. Congratulations! Your manuscript is now with our production department. 

Kind regards, 

on behalf of

Dr. Simon Clegg 

Academic Editor

PLOS ONE